# Semantic Evolvement Enhanced Graph Autoencoder for Rumor Detection

## ABSTRACT

Due to the rapid spread of rumors on social media, which can cause widespread harm to the Web and society, rumor detection has become an extremely important challenge. Recently, numerous rumor detection models have been proposed. They utilize textual information and the propagation structure of events. Some methods also introduce contrastive learning to enhance the robustness of model. However, these methods overlook the importance of semantic evolvement information of event in propagation process, which is often challenging to be truly learned in supervised training paradigms and those contrastive learning based rumor detection methods. To address this issue, we propose a novel semantic evolvement enhanced Graph Autoencoder for Rumor Detection (GARD) model in this paper. The model learns semantic evolvement information of events by capturing local semantic changes and global semantic evolvement information through specific graph autoencoder and reconstruction strategies. By combining semantic evolvement information and propagation structure information, the model achieves a comprehensive understanding of event propagation and perform accurate and robust detection, while also detecting rumors earlier by capturing semantic evolvement information in the early stages. Moreover, in order to enhance the model's ability to learn the distinct patterns of rumors and non-rumors, we introduce a uniformity regularizer to further improve the model's performance. Experimental results on three public benchmark datasets confirm the superiority of our GARD method over the state-of-the-art approaches in both overall performance and early rumor detection.

## CCS CONCEPTS

• **Computing methodologies** → **Machine learning**; • **Information systems** → **World Wide Web**.

## KEYWORDS

Rumor Detection, Graph Autoencoder, Social Media, Graph Representation

**ACM Reference Format:**
Anonymous Author(s). 2023. Semantic Evolvement Enhanced Graph Autoencoder for Rumor Detection. In *Proceedings of the ACM Web Conference 2024 (WWW '24)*. ACM, New York, NY, USA, 9 pages. https://doi.org/10.1145/nnnnnnn.nnnnnnn

## 1 INTRODUCTION

The growth of the Web and social media has accelerated the diffusion of news information, enabling real-time discussions. However, this development also poses certain risks, such as the spread of rumors that can reduce the credibility of information on the Web, affecting people's lives and the stability of society [5, 34, 39, 58]. Therefore, there is an urgent need for a rapid and effective rumor detection method, which has also become one of the objectives in the field of Web mining. Conventional methods use handcrafted features such as text content [4], user characteristics [52], and propagation patterns [17, 44] to train supervised rumor classifiers, such as decision trees and support vector machines. However, these traditional models usually rely on local features for classification, whereas in rumor detection, understanding the context and global information of the text is crucial [1]. Recently, deep learning has played a crucial role in rumor detection by automatically learning high-level representations of text and propagation structures of rumors [18]. Many deep learning models, such as Recursive Neural Networks (RvNNs), Recurrent Neural Networks (RNNs) and its successors including Long Short-Term Memory (LSTM) networks, have been applied to rumor detection due to their ability to learn sequential features [3, 6, 25, 26, 31, 46]. However, these methods overlook the importance of complex topological structural information in rumor propagation.

In order to address this issue, some studies have invoke Graph Neural Networks (GNNs) to model the complex topological structural information of rumor propagation [2, 30, 42, 54]. Despite these models based on GNNs achieve success in rumor detection by effectively exploiting the structure information of propagation graphs, they often struggle to learn the intrinsic relationships between posts, because they only rely on supervised training objectives. This limitation results in poor generalization ability and unsatisfactory performance in real-world scenarios [23, 53]. Thus, recent works such as GACL [40] have proposed supervised graph adversarial contrastive learning method to capture the invariance of events, and RECL [51] perform a contrastive learning training based on relation-level augmentation and event-level augmentation, in order to enhance the robustness and generalization of models.

However, **the success of these rumor detection methods that introduce contrastive learning heavily relies on complex data augmentation techniques,** which require continuous trial and error to determine [45, 56]. Because unreasonable data augmentation methods often introduce more noise, leading to adverse effects on the model and causing a degradation in its performance [48, 55]. **Additionally, these models lack attention to the semantic evolvement during news propagation.** Semantic evolvement refers to the gradual transformation of the comprehensive semantics of news (including source post and all replies) as user interactions such as comments, shares, and likes increase. These comments often present diverse viewpoints due to different perspectives and positions, which contribute to the

alteration of the semantic meaning of the news. For example: **(1)** In the spread of a rumor, there is often a situation where initially, the majority of comments express agreement with the source post, but after some time, a large number of debunking messages appear. Therefore, capturing such a signal of significant semantic changes before and after can effectively detect rumors. **(2)** In the spread of a rumor, a portion of the comments may question and present evidence contradicting the source post, leading to semantic evolvement repeatedly within these contradictions. In contrast, in the spread of a non-rumor, the comments usually focus more on in-depth analysis and discussion of the information rather than refutation [38]. Capturing the overall semantic evolvement information can help model identify semantic transformations, and the way to obscure the truth, thereby recognizing features of misinformation. Furthermore, during the early stages of event propagation, rumors often share significant similarities in their structure because there is typically limited commenting and interaction [9], making it challenging to distinguish them solely based on structural features. So capturing the semantic evolvement information during early propagation stages can also help in identifying rumors early and minimizing the harm caused by misinformation. Therefore, it is crucial to consider and understand the semantic evolvement, and strive to capture such information during news propagation [21]. However, in prior work, these supervised training paradigms and contrastive learning based rumor detection methods struggled to enable models to learn genuine semantic information [37, 53].

In order to achieve more generalized, rapid, and effective rumor detection without the need for complex data augmentation techniques, we propose a novel semantic evolvement enhanced Graph Autoencoder for Rumor Detection (GARD) model in this paper. **It introduces self-supervised semantic evolvement learning to acquire more generalized and robust representations through feature reconstruction training based on propagation paths, while also detecting rumors earlier by capturing semantic evolvement information in the early stages.** Specifically, GARD learn the semantic evolvement information from both local and global perspectives: **(1)** In order to capture the local semantic changes between tweets and their retweets, we utilize the features of parent nodes to reconstruct the features of their child nodes in the top-down directions as shown in Fig. 1a, and utilize the features of child nodes to reconstruct the feature of their parent nodes in the bottom-up directions as shown in Fig. 1b. **(2)** In order to capture broader information propagation paths and contextual relationships, and determine whether significant semantic changes has occurred during the propagation of news, we introduce a global semantic learning module to learn the semantic relationships between each node and its multi-hop neighboring nodes by conducting features random mask reconstruction on undirected propagation graph. It randomly masks a portion of the nodes' features, then the masked features are reconstructed by their multi-hop neighboring nodes. **(3)** Furthermore, rumors and non-rumors usually exhibit distinct propagation patterns, and the propagation patterns differ among various event topics [9]. Therefore, in order to enhance the model's ability to learn the distinct patterns of rumors and non-rumors, we introduce a uniformity regularizer [36, 47] to further improve the model's performance, which prefers the uniform distribution on the unit hypersphere by pulling away the distance between

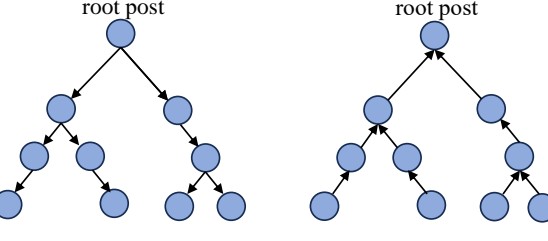

**(a) top-down direction**          **(b) bottom-up direction**

**Figure 1: (a) the top-down semantic evolvement graph, as comments increase, semantic begin to evolve, (b) the reverse bottom-top semantic evolvement graph, where the edges between nodes indicate the direction of features reconstruction.**

the representations of different events, so as to preserves maximal information and eliminates the features collapse issue [57].

The experimental results on three benchmark datasets demonstrate that our GARD outperforms state-of-the-art approaches in both overall performance and early rumor detection. The main contributions of our work are outlined as follows:

- We propose a GARD rumor detection method, which takes into account not only the structural features but also the crucial semantic evolvement features. This comprehensive consideration enables the model to achieve better robustness and generalization without the need for complex data augmentation techniques.
- In order to enhance the model's ability to learn distinctive propagation patterns of rumors and non-rumors, we introduce a uniformity regularizer that further improve the model's performance.
- Our GARD method has been evaluated on three widely used benchmark datasets, and the experimental results demonstrate its superiority over the state-of-the-art approaches in both overall performance and early rumor detection.

## 2 RELATED WORK

In this section, we briefly review prior work of rumor detection on social media, and introduce the current researches of graph autoencoders.

### 2.1 Rumor Detection

Most previous researches for rumor detection mainly based on machine learning methods. These conventional approaches involve using handcrafted features, such as text content [4], user characteristics [52], and propagation patterns [17, 44], to train supervised rumor classifiers like decision trees and support vector machines. In recent years, deep learning has emerged as a significant role in rumor detection by automatically learning high-level representations of text and rumor propagation structures. Various deep learning models, including RNN and its various variants, have been applied to rumor detection task [7, 22, 25, 26, 28, 32, 46]. To incorporate complex structural information into rumor propagation analysis, several approaches have incorporated GNNs to model the structural information within rumor propagation. By considering a more realistic representation of the problem, GNNs have demonstrated success in leveraging the structural information of propagation graphs [2, 24, 42, 43, 50].

To enhance the robustness and generalization of rumor detection models, some recent studies have proposed training methods that introduce supervised graph contrastive learning techniques to capture the invariance of events [40]. Some works also leverage unsupervised contrastive learning training methods to capture the repost relations and structural features of rumors [51]. But these works have lacked attention and exploration of semantic evolvement information, leading to a need for further improvement in the model's performance, especially in the early stages of event propagation, where the structural characteristics of events are similar.

## 2.2 Graph Autoencoders

Autoencoders [10] are designed to reconstruct certain inputs within a given context and do not impose any specific decoding order. The earliest works on Graph Autoencoders (GAEs) can be traced back to GAE and VGAE [13, 15], where they utilize a two-layer GCN as the encoder and employ dot-product for link prediction decoding. Later GAEs mostly adopted the structure reconstruction after VGAE or combined structure and feature reconstruction as their objectives [19, 33, 35, 41, 49]. In recent years, many studies have focused on investigating the effectiveness of masked feature reconstruction objectives for GNNs [14, 29, 53, 59]. Among them, GraphMAE [11, 12] has achieved good results in graph representation learning tasks based on masked feature reconstruction through the analysis of masking strategies and the design of loss functions. It has achieved state-of-the-art performance in multiple node classification and graph classification tasks.

## 3 PROBLEM DEFINITION

The problem of rumor detection is defined as a classification task, where the objective is to learn a classifier that can accurately detect rumors. Specifically, for a given rumor dataset $C = \{C_1, C_2, ..., C_M\}$, where $C_i$ is the $i$-th event and $M$ is the number of events. We defined each event $C_i = \{r, w_1, w_2, ..., w_{N_i-1}, \mathcal{G}_i\}$, where $N_i$ is the number of posts in $C_i$, $r$ refers to the source post, each $w_j$ represents the $j$-th relevant retweeted post or responsive post, and $\mathcal{G}_i$ defined as a graph represents the propagation structure of $C_i$. The graph $\mathcal{G}_i = \{V_i, A_i, X_i\}$, where $V_i$ refers to the set of nodes corresponding to $N_i$ posts and $A_i \in \{0, 1\}^{N_i \times N_i}$ as an adjacency matrix where:

$$a_{st}^i = \begin{cases} 1, & \text{if } e_{st}^i \in E_i \\ 0, & \text{otherwise,} \end{cases} \quad (1)$$

where $E_i = \{e_{st}^i | s, t \in \{0, 1, ..., N_i - 1\}\}$ represents the set of edges connecting a post to its retweeted posts or responsive posts as shown in Fig. 1a. $X_i = [x_0, x_1, ..., x_{N_i-1}]^T$ denote a node feature matrix extracted from the posts in $C_i$. We adopt the same approach as [40] by using the BERT [8] to separately encode the source and comments to form the feature matrix $X_i$. Besides, each event $C_i$ in the dataset is labeled with a ground-truth label $y_i$. Here, we define the problem statement as follows:

**Rumor Detection**: The task is to develop a classifier, denoted as $f : C_i \rightarrow y_i$, where $C_i$ represents a event of rumor dataset with their corresponding a graph structure and textual features.

## 4 THE PROPOSED GARD MODEL

In this section, we propose a GARD model for rumor detection tasks as Fig. 2 shows. GARD is mainly faced with two problems: (A) How to capture local semantic changes based on the propagation paths of events; (B) How to capture global semantic evolvement information based on the entire propagation structure of events. In response to the above problems, we will elaborate on the components of GARD, including local semantic evolvement learning, global semantic evolvement learning, representation of propagation graph, and uniformity regularizer.

## 4.1 Local Semantic Evolvement Learning

In order to capture the local semantic changes between tweets and their retweets, we proposed this Local Semantic Evolvement Learning module. We utilize the features of parent nodes to reconstruct the features of their child nodes in the top-down direction, and utilize the features of child nodes to reconstruct the features of their parent nodes in the bottom-up direction. In detail, given an input data $\mathcal{G} = (V, A, X)$ where $X \in \mathbb{R}^{N \times d}$, we obtain all $N_p$ parent-child node pairs, then obtain the parent feature matricx $X_p \in \mathbb{R}^{N_p \times d}$ for all parent nodes and child feature matricx $X_c \in \mathbb{R}^{N_p \times d}$ for all child nodes, respectively. Further, given $f_{local1}$ and $f_{local2}$ as two encoders, $g_{local1}$ and $g_{local2}$ as two decoders, here we use Multi-Layer Perceptron (MLP) as both the encoder and decoder. Then we individually input the parent feature matrix and child feature matrix into their respective encoder to obtain their latent representations. Next, we feed these representations into respective decoder to generate the reconstructed features. Formally, in the top-down direction, it can be written as follows:

$$H_p = f_{local1}(X_p), Z_p = g_{local1}(H_p), \quad (2)$$

in the bottom-up direction, it can be written as:

$$H_c = f_{local2}(X_c), Z_c = g_{local2}(H_c), \quad (3)$$

where $H_p, H_c \in \mathbb{R}^{N_p \times d_h}$ are the latent representations of parent nodes and child nodes, $Z_p, Z_c \in \mathbb{R}^{N_p \times d}$ is the reconstructed features. Then, we calculate the Mean Squared Error (MSE) loss between the original features and the reconstructed features in both top-down and bottom-up directions:

$$\mathcal{L}_{rec1} = \frac{1}{N_p}\frac{1}{d}\sum_{i=1}^{N_p}\sum_{j=1}^{d}(x_{ij}^c - z_{ij}^p)^2 + \frac{1}{N_p}\frac{1}{d}\sum_{i=1}^{N_p}\sum_{j=1}^{d}(x_{ij}^p - z_{ij}^c)^2, \quad (4)$$

where $x_{ij}^c$ and $z_{ij}^c$ refers to the j-th feature value of the i-th node in feature matrix $X_c$ and $Z_c$. The parameters of $f_{local1}$ and $f_{local2}$ can be learned by:

$$\Theta_1^{\star} = \arg\min_{\Theta_1} \mathcal{L}_{rec1}(\mathcal{G}; \Theta_1), \quad (5)$$

where $\Theta_1$ denotes the parameters of $f_{local1}$ and $f_{local2}$.

## 4.2 Global Semantic Evolvement Learning

In order to capture broader information propagation path and contextual relationships, to determine whether significant semantic changes has occurred during the propagation of news, we proposed this Global Semantic Evolvement Learning module. In detail, given an input data $\mathcal{G} = (V, A, X)$ where $X \in \mathbb{R}^{N \times d}$, we first apply a

**Figure 2: The overall framework of our proposed GARD model. Given a batch of input data, we perform both local and global semantic evolvement learning. (1) The learning of local semantic changes is achieved by reconstructing node features in both the top-down and bottom-up directions of parent-child node pairs. (2) The learning of global semantic evolvement is achieved by conducting features random mask reconstruction on undirected propagation graph. (3) We introduce a uniformity regularizer to enhance the model's ability to learn the distinctive patterns of events. The features reconstructed loss, supervised loss, and the uniformity loss are combined to update the model parameters.**

uniform random sampling strategy with a mask ratio to sample a subset of nodes $\widetilde{V} \in V$ and mask each of their features with a mask token [MASK], i.e., a learnable vector $x_{[MASK]} \in \mathbb{R}^d$. Thus, the node feature $\tilde{x}_i$ for $v_i \in V$ in the masked feature matrix $\widetilde{X}$ can be defined as:

$$\tilde{x}_i = \begin{cases} x_{[MASK]} & v_i \in \widetilde{V} \\ x_i & v_i \notin \widetilde{V}. \end{cases} \qquad (6)$$

Further, given $f_{global}$ as an encoder and $g_{global}$ as a decoder, here we use Graph Convolutional Network (GCN) [16] as both the encoder and decoder, in which, each node relies on its neighbor nodes to enhance/recover features. Then we take the obtained feature matrix $\widetilde{X}$ and adjacency matrix $A$ as inputs to the encoder to obtain latent representations. Next, these representations are fed into the decoder to generate the reconstructed feature matrix. Formally, it can be written as follow:

$$H = f_{global}(A, \widetilde{X}), Z = g_{global}(A, H), \qquad (7)$$

where $H \in \mathbb{R}^{N \times d_h}$ is the latent representations of input nodes, $Z \in \mathbb{R}^{N \times d}$ is the reconstructed features. Then, we calculate the MSE loss between the original features and the reconstructed features of the masked nodes:

$$\mathcal{L}_{rec2} = \frac{1}{N_m} \frac{1}{d} \sum_{i=1}^{N_m} \sum_{j=1}^{d} (x_{ij} - z_{ij})^2, \qquad (8)$$

where $N_m$ represents the number of masked nodes. Please note that we only calculate the MSE loss on the masked node features. The parameters of encoder $f_{global}$ can be learned by:

$$\Theta_2^\star = \arg\min_{\Theta_2} \mathcal{L}_{rec2}(\mathcal{G}; \Theta_2), \qquad (9)$$

where $\Theta_2$ denotes the parameters of $f_{global}$.

### 4.3 Representation of Propagation Graph

In order to leverage label information, we also calculate a supervised loss function for optimizing the model. Specifically, given an input data $\mathcal{G} = (V, A, X)$ where $X \in \mathbb{R}^{N \times d}$, we input the data into encoder $f_{local1}$, $f_{local2}$ and $f_{global}$ to obtain latent representations, respectively. Then, we use mean-pooling operators ($MEAN$) to aggregate the information of the set of node representations. Finally, we concatenate them to merge the information. Formally, it can be written as follow:

$$H_{k1} = f_{local1}(X), H_{k2} = f_{local2}(X), H_j = f_{global}(A, X), \qquad (10)$$

$$h_{k1} = MEAN(H_{k1}), h_{k2} = MEAN(H_{k2}), h_j = MEAN(H_j), \qquad (11)$$

$$m = concat(h_{k1}, h_{k2}, h_j), \qquad (12)$$

where $m \in \mathbb{R}^{3d_h}$ denotes the representation of event. Next, $m$ is fed into full-connection layers and a softmax layer, and the output is calculated as:

$$\hat{y} = softmax(W_k m + b_k), \qquad (13)$$

where $\hat{\boldsymbol{y}} \in \mathbb{R}^C$ is a vector of probabilities for all the classes $C$. $\boldsymbol{W}_k \in \mathbb{R}^{C \times 3d_h}$ and $\boldsymbol{b}_k \in \mathbb{R}^C$ are the learnable weight matrix and bias respectively.

Therefore, we introduce a cross-entropy as supervised loss into the objective of encoder $f_{local1}$, $f_{local2}$ and $f_{global}$. The objective are updated as:

$$\mathcal{L} = \mathcal{L}_{\text{sup}}(\boldsymbol{G};\Theta_1,\Theta_2) + \alpha(\mathcal{L}_{\text{rec1}}(\boldsymbol{G};\Theta_1) + \mathcal{L}_{\text{rec2}}(\boldsymbol{G};\Theta_2)), \quad (14)$$

where

$$\mathcal{L}_{\text{sup}} = -\frac{1}{N}\sum_{k=1}^{N}\sum_{j=1}^{C}\boldsymbol{y}_{k,j}log(\hat{\boldsymbol{y}}_{k,j}), \quad (15)$$

and $\alpha$ is an adjustable hyperparameter used to control the weight of the reconstructed loss. In $\mathcal{L}_{\text{sup}}$, $\boldsymbol{y}_{k,j}$ denotes ground-truth label that has been one-hot encoded. and $\hat{\boldsymbol{y}}_{k,j}$ denotes the predicted probability distribution of event index $k \in \{1, 2..., N\}$ belongs to class $j \in \{1, 2, ...C\}$.

During the testing phase, we do not perform any special processing on the input data. We simply input it into all encoders to obtain their representations like Eqs. (10) to (13) to generate the classification predictions.

## 4.4 Uniformity Regularizer

In order to enhance the model's ability to learn the distinct patterns of rumors and non-rumors, we introduce a uniformity regularizer to further improve the model's performance. Uniformity loss prefers the uniform distribution on the unit hypersphere by pulling away the distance between the representations of different events, so as to preserves maximal information and eliminates the feature collapse issue [57]. The uniformity loss is defined as the logarithm of the average pairwise Gaussian potential:

$$\mathcal{L}_{\text{uni}} = \log \mathop{\mathbb{E}}_{(\boldsymbol{m}_i,\boldsymbol{m}_j)\sim p_{\text{data}}} e^{-t\left\|\boldsymbol{m}_i-\boldsymbol{m}_j\right\|^2}, \quad (16)$$

where $p_{data}$ is the distribution of data, $t$ is a hyperparameter for Gaussian potential kernel and $\boldsymbol{m}_k$ denotes the graph representations of event $k$.

Then, we introduce a uniformity loss into the objective of encoder $f_{local1}$, $f_{local2}$ and $f_{global}$. The objective defined by Eq. (14) are finally updated as:

$$\begin{aligned}\mathcal{L} = \mathcal{L}_{\text{sup}}(\boldsymbol{G};\Theta_1,\Theta_2) + \alpha_1(\mathcal{L}_{\text{rec1}}(\boldsymbol{G};\Theta_1) \\ + \mathcal{L}_{\text{rec2}}(\boldsymbol{G};\Theta_2)) + \alpha_2\mathcal{L}_{\text{uni}}(\boldsymbol{G};\Theta_1,\Theta_2),\end{aligned} \quad (17)$$

where $\alpha_1$ and $\alpha_2$ are adjustable hyperparameters used to control the weight of the reconstructed loss and uniformity loss.

The parameters updating defined by Eq. (5) and Eq. (9) are updated as:

$$\Theta_1^{\star},\Theta_2^{\star} = \arg\min_{\Theta_1,\Theta_2}\mathcal{L}(\boldsymbol{G};\Theta_1,\Theta_2), \quad (18)$$

To help better understand the training process of GARD, we provide the brief pseudo-code of it in Algorithm 1.

## 5 EXPERIMENTS

In this section, we first conduct experiments to evaluate the effectiveness of the proposed GARD model by comparing it with other baseline models for rumor detection, and give some discussion and analysis. Secondly, we conducted ablation study to evaluate and analyze the effectiveness of each module in GARD. Thirdly, we

---

**Algorithm 1:** Training process of GARD

**Input** : A set of input graphs $\boldsymbol{G}$, maxEpoch

1 Initialize $\Theta_1, \Theta_2$ with random weight values.
2 **for** *epoch from 1 to maxEpoch* **do**
3      **for** *each mini-batch of $\boldsymbol{G}$* **do**
4          Construct all parent-child node pairs.
5          Reconstruct node features using Eqs. (2) and (3).
6          Compute local reconstructed loss using Eq. (4).
7          Randomly mask a portion of nodes' features on undirected graph.
8          Reconstruct node features using Eq. (7).
9          Compute global reconstructed loss using Eq. (8).
10         Calculate the graph representation using Eqs. (10) to (12).
11         Compute supervised loss using Eq. (15).
12         Compute uniformity loss using Eq. (16).
13         Compute total loss using Eq. (17).
14         Update $\Theta_1$ and $\Theta_2$ with the gradient of Eq. (18).
15      **end for**
16 **end for**

---

perform a sensitivity analysis of the hyper-parameters in GARD, discussing the impact of each hyper-parameter on the experimental results. Finally, we evaluate the performance of GARD in the task of early rumor detection.

### 5.1 Evaluation Setups

*5.1.1 **Datasets**.* We conducted an evaluation of the GARD model using three publicly available real-world datasets: Twitter15 [27], Twitter16 [27], and PHEME [20]. These datasets were collected from Twitter, which is considered the most influential social media site in the US. The PHEME dataset consists of two versions based on five and nine breaking news events, and we selected the version with nine events for our work. Both Twitter15 and Twitter16 datasets have four tags: Non-rumor (N; Confirmed to be true), False Rumor (F; Confirmed to be a rumor), True Rumor (T; Initially thought to be a rumor but later confirmed to be true), and Unverified Rumor (U; The truthfulness is yet to be determined). The PHEME dataset only has two tags: Rumor (R) and Non-Rumor (N), used for binary classification of rumors and non-rumors. For detailed statistics, please refer to Table 1.

*5.1.2 **Baselines**.* We compare GARD with SOTA rumor detection models, including:

- **DTC** [4]: A rumor detection method employs a Decision Tree classifier to detect rumors by analyzing a set of handcrafted features.
- **SVM-TS** [26]: A method utilizes a linear SVM classifier and handcrafted features to build a time-series model.
- **BERT** [8]: A popular pre-trained model that is used for rumor detection.
- **RvNN** [28]: A rumor detection approach based on tree-structured recursive neural networks with GRU units that learn rumor representations via the propagation structure.

**Table 1: Statistics of the datasets**

| Statistics | $Twitter15$ | $Twitter16$ | $PHEME$ |
|---|---|---|---|
| # source posts | 1490 | 818 | 6425 |
| # non-rumors | 374 | 205 | 4023 |
| # false rumors | 370 | 205 | 2402 |
| # unverified rumors | 374 | 203 | - |
| # true rumors | 372 | 205 | - |
| # users | 276,663 | 173,487 | 48,843 |
| # posts | 331,612 | 204,820 | 197,852 |

- **GCAN** [24]: A GNN-based model that can describe the rumor propagation mode and use the dual co-attention mechanism to capture the relationship between source text, user characteristics and propagation path.
- **BiGCN** [2]: A GNN-based rumor detection model utilizing the Bi-directional propagation structure.
- **GACL** [40]: A GNN-based model using adversarial and contrastive learning, which can not only encode the global propagation structure, but also resist noise and adversarial samples, and captures the event invariant features by utilizing contrastive learning.
- **RECL** [51]: A rumor detection model perform contrastive learning at both the relation level and event level to enrich the self-supervision signals for rumor detection.
- **GARD (ours)**: A rumor detection model introduces self-supervised semantic evolvement learning to facilitate the acquisition of more transferable and robust representations.

*5.1.3* ***Experimental Settings***. We follow the evaluation protocol in BIGCN[2]. We randomly split the dataset into five parts and construct 5-fold cross-validation. The Accuracy (Acc.), Precision (Prec.), Recall (Rec.) and $F1$-measure ($F1$) are adopted as evaluation metrics in all three datasets. Same as GACL [40], graph topologies of posts are constructed based on users, sources and comments in the all three datasets, where the text content contained in each graph node is represented by BERT. Furthermore, the learning rate is set to $5e - 4$ and the mask ratio in global semantic evolvement learning module is set to 0.25. We adopt 2-layer MLP as backbone of two encoders $f_{local}$ and tow decoders $g_{local}$, while adopt 2-layer GCN as encoder $f_{global}$ and 1-layer GCN as decoder $g_{global}$. We set $\alpha_1 = 0.05, \alpha_2 = 0.5$ for Twitter15 and Twitter16, and $\alpha_1 = 0.1, \alpha_2 = 1$ for PHEME.

## 5.2 Overall Performance

Table 2 and Table 3 show the performance of the proposed GARD and all the compared methods on three public real-world datasets, where the bold part represents the best performance. The experimental results demonstrate that the proposed GARD performs exceptionally well among all baseline models, confirming the advantages of incorporating Graph Autoencoder to learn the semantic evolvement information of news propagation.

Not surprisingly, the machine learning-based models, DTC and SVM-TS, obtained the worst results. On the other hand, the deep learning-based models, RvNN and BERT, achieved moderate performance in the tests. Both GCAN and BiGCN are models based on GNN. They relied on a powerful GNN encoder to capture global

**Table 2: Rumor detection results on Twitter15 and Twitter16 datasets (N: Non-Rumor; F: False Rumor; T: True Rumor; U: Unverified Rumor)**

| | | N | F | T | U |
|---|---|---|---|---|---|
| **$Twitter15$** | | | | | |
| Model | Acc. | $F1$ | $F1$ | $F1$ | $F1$ |
| DTC | 0.454 | 0.415 | 0.355 | 0.733 | 0.317 |
| SVM-TS | 0.642 | 0.811 | 0.434 | 0.639 | 0.600 |
| RvNN | 0.723 | 0.682 | 0.758 | 0.821 | 0.654 |
| BERT | 0.735 | 0.731 | 0.722 | 0.730 | 0.705 |
| GCAN | 0.842 | 0.844 | 0.846 | 0.889 | 0.800 |
| BIGCN | 0.886 | 0.891 | 0.860 | 0.930 | 0.864 |
| GACL | 0.901 | **0.958** | 0.851 | 0.903 | 0.876 |
| RECL | 0.902 | 0.856 | 0.910 | **0.947** | 0.894 |
| GARD | **0.911** | 0.889 | **0.923** | 0.905 | **0.901** |
| **$Twitter16$** | | | | | |
| Model | Acc. | $F1$ | $F1$ | $F1$ | $F1$ |
| DTC | 0.473 | 0.254 | 0.080 | 0.190 | 0.482 |
| SVM-TS | 0.691 | 0.763 | 0.483 | 0.722 | 0.690 |
| RvNN | 0.737 | 0.662 | 0.743 | 0.835 | 0.708 |
| BERT | 0.804 | 0.777 | 0.525 | 0.824 | 0.787 |
| GCAN | 0.871 | 0.857 | 0.688 | 0.929 | 0.901 |
| BIGCN | 0.880 | 0.847 | 0.869 | 0.937 | 0.865 |
| GACL | 0.920 | 0.934 | 0.869 | **0.959** | 0.907 |
| RECL | 0.921 | 0.875 | 0.933 | 0.949 | 0.901 |
| GARD | **0.932** | **0.936** | **0.935** | 0.950 | **0.908** |

structural features of the rumor tree. By integrating the bottom-up and top-down structural information of rumors, BiGCN achieved a great increase in average accuracy on three datasets. GACL and RECL are both models based on GNN and contrastive learning, which improve the model's robustness through specific data augmentation strategies and contrastive learning methods. They serve as state-of-the-art benchmarks to validate the advantages of the proposed GARD model in this paper.

The GARD model proposed in this paper achieved the best performance on all benchmarks, because with the progress of information propagation, particularly in the case of larger data volumes and higher data quality, there is a greater possibility of significant semantic changes in news' propagation and have more data allows the model to learn a greater amount of semantic knowledge. Therefore, learning semantic evolvement information becomes more important. Paying attention to it helps improve the performance of rumor detection tasks, and so our GARD model achieved the best performance without the need for complex data augmentation strategies.

Additionally, we found that the accuracy on the PHEME dataset is relatively lower compared to Twitter. This is because the PHEME dataset consists of data from only 9 event topics, leading to a significant overlap in the language descriptions and propagation structure. And our GARD achieve more improvement on the PHEME dataset

**Table 3: Rumor detection results on PHEME dataset**

| Method | Class | Acc. | Prec. | Rec. | F1 |
|--------|-------|------|-------|------|-----|
| | | | *PHEME* | | |
| DTC | R | 0.254 | 0.080 | 0.190 | 0.482 |
| | N | | 0.483 | 0.722 | 0.690 |
| SVM-TS | R | 0.685 | 0.553 | 0.539 | 0.539 |
| | N | | 0.758 | 0.762 | 0.757 |
| RvNN | R | 0.763 | 0.689 | 0.587 | 0.631 |
| | N | | 0.796 | 0.858 | 0.825 |
| BERT | R | 0.807 | 0.736 | 0.695 | 0.713 |
| | N | | 0.842 | 0.866 | 0.853 |
| GCAN | R | 0.834 | 0.769 | 0.758 | 0.761 |
| | N | | 0.871 | 0.874 | 0.872 |
| BIGCN | R | 0.824 | 0.753 | 0.734 | 0.741 |
| | N | | 0.861 | 0.872 | 0.865 |
| GACL | R | 0.850 | 0.801 | 0.750 | 0.772 |
| | N | | 0.871 | 0.901 | 0.885 |
| RECL | R | 0.852 | 0.800 | 0.753 | 0.778 |
| | N | | 0.868 | 0.910 | **0.888** |
| **GARD** | R | **0.869** | **0.817** | **0.764** | **0.790** |
| | N | | **0.886** | **0.928** | 0.886 |

than Twitter dataset because our model takes into account not only the structural information but also the crucial semantic evolvement information which exhibits greater distinctiveness on the PHEME dataset.

## 5.3 Ablation Study

To evaluate the efficacy of the various modules of GARD, we conduct a comparative analysis by comparing it with the following variants:

- **GARD-SUP**: This model removes the two semantic evolvement learning modules and the uniformity regularizer, and solely conducts supervised training by inputting a complete propagation graph into two encoders.
- **GARD-NGS**: This model removes the global semantic learning module, which makes the model lose the ability of capturing broader significant semantic evolvement during the propagation of news.
- **GARD-NLS**: This model removes the local semantic learning module, which makes the model lose the ability of capturing the local semantic changes between tweets and their retweets in both the top-down and bottom-up propagation directions.
- **GARD-NU**: This model removes the uniformity regularizer, which makes the model lose the ability of eliminating the features collapse issue, allowing the model to learn more uniform representations.

The results are summarized in Table 4. We have the following observations from this table:

1) By comparing GARD and GARD-SUP (also can compare GARD-NU and GARD-SUP), we can observe that the accuracy of GARD-SUP on the Twitter15, Twitter16 and PHEME datasets

**Table 4: Results of ablation study on the Twitter15, Twitter16 and PHEME**

| Model | Acc. | | |
|-------|------|------|------|
| | *Twitter*15 | *Twitter*16 | *PHEME* |
| GARD | **0.911** | **0.932** | **0.869** |
| GARD-SUP | 0.862 | 0.875 | 0.822 |
| GARD-NGS | 0.894 | 0.913 | 0.843 |
| GARD-NLS | 0.895 | 0.901 | 0.850 |
| GARD-NU | 0.905 | 0.926 | 0.861 |

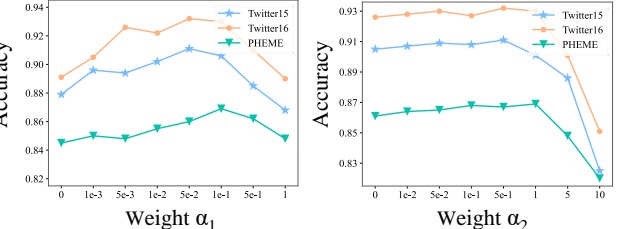

**Figure 3: Sensitivity analysis of hyperparameters $\alpha_1$ and $\alpha_2$, which represent the weight of reconstructed loss and uniformity loss. The y-axis represents accuracy(%) and the x-axis is the different hyper-parameters.**

is reduced by 4.9%, 5.7% and 4.7%, respectively. Obviously, the introduction of self-supervised semantic evolvement learning in our GARD leads to significant performance improvement compared to solely using a supervised learning objective to train the model.

2) Removing either the local semantic evolvement learning module or the global semantic evolvement learning module results in a decrease in the model's performance, but both perform better than GARD-SUP, which includes no semantic learning module. The best performance is achieved when both modules are present together, which demonstrates that both local and global semantic evolvement learning modules are beneficial and the combination of local semantic evolvement information and global semantic evolvement information provides a greater improvement.

3) By comparing GARD and GARD-NU, we can observe that the uniformity regularizer improves the performance of the model to a certain extent. In particular, it increased by 0.8% on the PHEME dataset. This is because the PHEME dataset has only 9 event topics, which results in a more similar event propagation structure and language description compared to Twitter. The uniformity enhances the model's ability to learn distinguishing features, leading to a more significant improvement.

## 5.4 Sensitivity Analysis

We conduct sensitivity analysis with hyper-parameters on the key designs of GARD. Fig. 3 shows the effect of varied hyper-parameter values, from which we have the following observations.

*5.4.1 **Effect of weight of reconstruction loss** $\alpha_1$.* This weight affects the result of rumor detection by affecting the weight of reconstruction loss in the total loss. As shown in the left picture of Fig. 3,

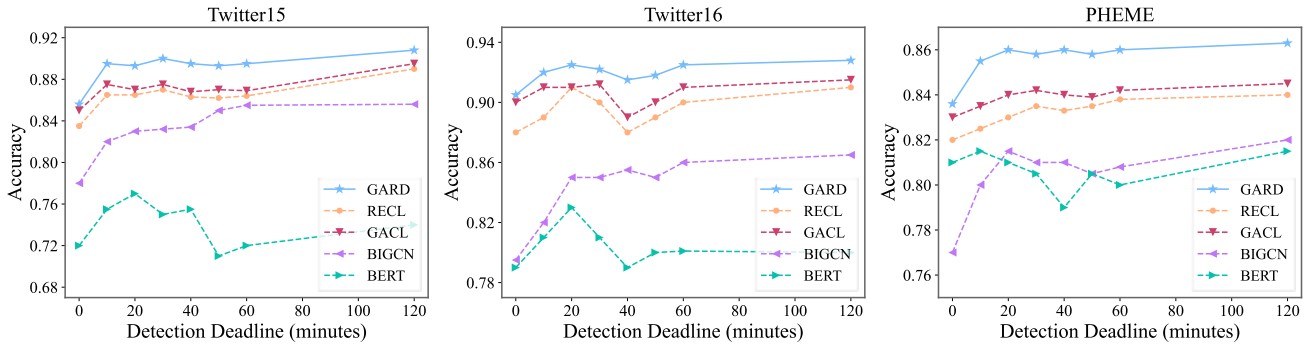

**Figure 4: Results of rumor early detection task on three datasets**

we conducted sensitivity analysis by selecting eight data points between 0 and 1. It can be observed that as the hyper-parameter $\alpha_1$ gradually increases, the model's performance on three datasets starts to improve due to graph autoencoder self-supervised learning allows the model to learn semantic evolvement information, thereby improving the model's performance. Although there are some fluctuations at certain positions, overall performance improves with the increase of $\alpha_1$. For Twitter15 and Twitter16 datasets, the best performance is achieved when $\alpha_1$ is set to $5e-2$, while for the PHEME dataset, $\alpha_1$ of $1e-1$ yields the best performance. It is worth noting that when $\alpha_1$ exceeds a certain threshold, the model's performance starts to decline noticeably. This is because of overfitting of the model to the self-supervised features reconstruction task during training.

*5.4.2* **Effect of weight of uniformity loss** $\alpha_2$**.** This weight affects the result of rumor detection by affecting the weight of uniformity loss in the total loss. As shown in the right picture of Fig. 3, we conducted sensitivity analysis by selecting eight data points between 0 and 10. We can observe that initially, as $\alpha_2$ increases, the model's performance on Twitter shows slow improvement. The performance improvement on Twitter is relatively stable, but overall, it improves with the increase of $\alpha_2$. However, on the PHEME dataset, there is a more significant improvement in model performance as $\alpha_2$ increases. This is due to the fact that, as mentioned earlier, the PHEME dataset has a smaller number of event topics, making the value of $\alpha_2$ have a larger impact on performance. The model achieves the best performance on Twitter15 and Twitter16 when $\alpha_2$ is set to $5e-1$, and on the PHEME dataset, the best performance is achieved when $\alpha_2$ is set to 1. Similarly, when $\alpha_2$ exceeds a certain threshold, the model's performance starts to decline noticeably. This is because excessively pursuing the learning of feature differences can actually harm the quality of the learned representations, leading to a decrease in classification accuracy.

## 5.5 Early Rumor Detection

Early rumor detection is also an important way for evaluating models. Its purpose is to detect rumors during the early stages of their spread, thereby preventing potentially greater harm. In our experiments in this paper, similar to the [40], we set up 8 different time points (i.e., 10, 20, ..., 120 minutes) to evaluate whether the model can correctly identify rumors based on the limited information available from earlier time points up to these specific moments.

Fig. 4 shows the performances of our GARD versus RECL, GACL, BIGCN and BERT at various deadlines for the Twitter15, Twitter16 and PHEME datasets in the early rumor detection task. We can observe that at time 0, all the models perform poorly. This is because at this stage, only the source post exists, and the crucial clue of comment information is missing, which enables the model to better detect rumors. But at 10 minutes, our GARD model shows a more significant improvement compared to other models, and it maintains a high and stable accuracy rate throughout the subsequent time periods. This is because in the early stages of event propagation, there is less commenting and interaction, resulting in similar propagation structures for events. Therefore, relying solely on structural information to detect rumors has significant limitations. Our GARD model, on the other hand, not only considers structural information but also takes into account the semantic evolvement information. This comprehensive understanding allows the model to effectively detect rumors in the early stages. The performances demonstrates that semantic evolvement information are not only beneficial to long-term rumor detection, but also helpful to the early detection of rumors.

## 6 CONCLUSION

In this paper, we propose a novel rumor detection model GARD, which detects rumors by effectively introducing self-supervised semantic evolvement learning to facilitate the acquisition of more transferable and robust representations through feature reconstruction training based on propagation paths, while also detecting rumors earlier by capturing semantic evolvement information in the early stages. Our model learn local semantic changes based on propagation paths effectively by using the parent nodes to reconstruct the features of their child nodes in the top-down direction and utilizing child nodes to reconstruct the features of their parent nodes in the bottom-up direction. And it capture global semantic evolvement information based on propagation structure by conducting a random masked features reconstruction on undirected graph. Additionally, we have introduced a uniformity regularizer to further enhance the model's performance. By comprehensively capturing the semantic evolvement information and structure information of events, our proposed GARD method consistently outperforms existing state-of-the-art methods in both overall performance and early rumor detection.

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
