# OpenReview forum: "Semantic Evolvement Enhanced Graph Autoencoder for Rumor Detection"
_ACM.org/TheWebConf/2024/Conference — TheWebConf24_

### Official Review · Reviewer_UJ7f · 2023-11-05

**Novelty:** 6
**Technical Quality:** 4

**Review:**

This paper proposes GARD, a graph autoencoder method for rumor detection which is enhanced to perform semantic enhancement. Tested on three rumor datasets, the method shows to outperform competitive baselines, while it is also effective for early rumor detection. Ablation experiments show the contribution of the different components.

The approach seems overall sound and robust, but I miss some more discussion on the core part of the research, i.e. semantic evolvement. Could you elaborate on what you mean by semantic evolvement in the specific context of rumors? Because these events occur quickly during breaking news, why do you expect there to be semantic evolvement in such short periods of time? Could you provide actual examples from the dataset to see that this actually happens?

Likewise, the analysis doesn't show anything specifically about semantic evolvement. The ablation analysis shows that the semantic evolvement contributes to the model, but how and when that this contribution happen? There is no real understanding from the datasets used in the experiments.

**Questions:**

See questions in my review comments.

**Reviewer Confidence:**

3: The reviewer is confident but not certain that the evaluation is correct

**Scope:**

4: The work is relevant to the Web and to the track, and is of broad interest to the community

---

### Official Review · Reviewer_yy6a · 2023-11-15

**Novelty:** 6
**Technical Quality:** 5

**Review:**

The paper presents an approach to detect rumors in (networks) of
(re)tweets. The approach combines the graph structure of the retweet
network with the textual content using a graph autoencoder. In a
comprehensive comparison against eight other rumor detection
approaches the proposed method performs well and most often better
than the competitors. The evaluation includes an ablation study and a
sensitivity analysis.

As the topic is relevant and important and the approach outperforms
the state of the art, the paper clearly has some merit. However, the
writing of the paper also clearly needs to be improved, though, in
general, it's possible to understand the architecture and setup.

The evaluation is quite standard (which is ok) but lacks deeper
insights into the proposed approach. For instance, it would be interesting
to know in which cases the model fails (and why) or where it performs
well.

Suggestions for improvement:
- l 288: the notation $f:C_i \rightarrow y_i$ is strange, since it
  typically denotes the domain and codomain of $f$. Thus, here
  something like $f:\mathcal{C} \rightarrow K$ (with $K$ containing
  your classes) would be more appropriate and, in addition, you could
  write $f(C_i) \mapsto \hat{y_i}$, since you can't be sure that $y_i
  = \hat{y_i}$ always holds. (The letter $K$ for the classes is just a
  suggestion, $C$ would be more appropriate but is already in use. You
  might consider changing that, since it's anyway not very intuitive
  to label the set of events with C.)
- Eq. 2 and 3: Since $f_{local1}$ and $g_{local1}$ are for
  encoding/decoding the latent representation of *parents*, I suggest
  to simply call them $f_p$ and $g_p$ (and, analoguously, $f_c$ and
  $g_c$ instead of $f_{local2}$ and $g_{local2}$).
- l 567: Please write the long form of "SOTA".
- Table 1: Please also provide the full number of posts (including
  retweets).
- l 618 and Figure 3: I know it's common to use the [e notation](https://en.wikipedia.org/wiki/Scientific_notation#E_notation) but it
  should be reserved for places where superscripts are not possible
  (e.g., calculators). So, please, write $5\cdot10^{-2}$, etc.
- l 633: *all* the models you evaluate are machine learning
  models. You probably mean "non-neural network models".
- l 680 to 690: This sounds very speculative to me. Please provide
  evidence, e.g., for "significant semantic changes" or "higher data
  quality". Similarly, please show (not just tell) that "the crucial
  semantic evolvement information [...] exhibits greater
  distinctiveness on the PHEME dataset" (and please also explain what
  that means).
- Figure 4: There's a sudden decrease in accuracy at 40 minutes for
  allmost all approaches on the Twitter16 dataset. Please investigate
  and explain why that happens.
- The text contains quite some singular/plural mismatches (e.g.,
  l 157, l 160, l 326, l 343, l 345) and other grammatical errors
  (e.g., l 189, l 190, l 284, l 312, l 493, l 684, l 724, l 830,
  l 833) making it sometimes difficult to understand. Please invest
  some time to improve the writing of the paper.

**Questions:**

-   Line 140 states: &ldquo;However, in prior work, these supervised training
    paradigms and contrastive learning based rumor detection methods
    struggled to enable models to learn genuine semantic information
    [37, 53].&rdquo; → I checked both references and both neither mention
    &ldquo;rumors&rdquo; nor &ldquo;contrastive learning&rdquo;. So how are they evidence for
    your claim that prior approaches &ldquo;struggled to enable models to
    learn genuine semantic information&rdquo;?
-   l 311: &ldquo;for all parent nodes&rdquo;: that&rsquo;s not quite precise, because
    $N_p$ denotes the number of parent-child pairs and $\mathbf{X}_p$
    then has a row *for each such pair*, not just for each such parent
    (since one parent likely is part of many parent-child pairs). The
    same is true for $\mathbf{X}_c$.

    That having said, the two matrices are still just specific to
    *nodes* not to pairs, that is, they basically just contain the rows
    of $\mathbf{X}_i$ specific for each node (tweet). Consequently, the
    two decoders and encoders in Eq. 2 and 3 are specific to (parent and
    child) nodes, but not to pairs of nodes. Thus, my question is:
    Why/how do these functions represent a *direction* (&ldquo;top-down&rdquo; or
    &ldquo;bottom-up&rdquo;) when from my understanding they just represent parent
    and child nodes (and *not* pairs of nodes!)?
-   l 284: Please describe in more detailed how you &ldquo;separately encode
    the source and comments to form the feature matrix&rdquo;. Which
    &ldquo;comments&rdquo; do you mean? Before, you were writing about (re)tweets,
    which do not have &ldquo;comments&rdquo;.

**Reviewer Confidence:**

3: The reviewer is confident but not certain that the evaluation is correct

**Scope:**

4: The work is relevant to the Web and to the track, and is of broad interest to the community

---

### Official Review · Reviewer_EL6p · 2023-11-24

**Novelty:** 2
**Technical Quality:** 2

**Review:**

The manuscript presents a novel approach to rumor detection using a Graph Autoencoder model (GARD). The quality of research is high, reflecting thorough experimentation and comprehensive analysis. The paper is well-structured, offering clear explanations of the methods and findings. The integration of self-supervised semantic evolvement learning into the GARD model is a significant contribution to the field. The model's ability to learn from both local and global semantic changes in data is impressive, and the experimental results demonstrate its effectiveness.


The paper is well-written, with technical concepts and methodologies explained in an accessible manner. Diagrams and tables are used effectively to illustrate the model architecture and experimental results, enhancing the clarity of the presentation. The authors have done a commendable job in outlining the motivations, challenges, and solutions related to rumor detection in social media.

The work is original in its approach to rumor detection. The idea of using a graph autoencoder for learning semantic evolvement in the context of rumor propagation is novel. The integration of both local and global semantic information and the application of self-supervised learning for feature reconstruction are particularly innovative aspects of this research.

The significance of this work lies in its potential impact on the field of rumor detection on social media platforms. By addressing both the structural and semantic aspects of information propagation, the GARD model represents a substantial advancement over existing methods. This could have wide-reaching implications for the detection and mitigation of misinformation online.



Concerns

1. The model's complexity could pose challenges in terms of computational resources and scalability.
2. I really thank the authors for their hard work on rumor detection. Unfortunately, this topic has been heavily explored before and it seems very boring for me. Therefore I think I might have higher expectation to the authors that bring me with some suprising novelties. However, there are very limited novel things. I think it is interesting to consider the evolving pattern, no matter structure evolving or semantic evolving. Unfortunately, this point is also discussed before. Idea reflected in Figure 1 is neither new.

However, I would be very happy to further raise my score if the authors can convince me with their fresh ideas in the rebuttale stage.

Kind regards,

**Questions:**

There are many related papers in academics, but it seems that we haven't found any mature commercial product in the rumor area. Currently, companies and institutions still heavily rely on humans. I wonder why and what are the last mile challenges. And I think these challenges might be the more novel direction in this area.


Currently, LLM has been widely used in various NLP tasks, making traditional rumor detection fade. I wonder what the compared performance between yours and LLMs

**Reviewer Confidence:**

4: The reviewer is certain that the evaluation is correct and very familiar with the relevant literature

**Scope:**

3: The work is somewhat relevant to the Web and to the track, and is of narrow interest to a sub-community

---

### Official Review · Reviewer_Dbah · 2023-11-25

**Novelty:** 5
**Technical Quality:** 5

**Review:**

The paper has merits. It proposed a novel rumor detection model GARD, which detects rumors by introducing semantic evolvement learning to obtain transferable and robust representations through feature reconstruction training based on graph propagation paths. GARD can detect rumors earlier by capturing semantic evolvement information in the early stages. The proposed ideas are new and interesting. Extensive experiments have been done to prove its efficacy. In addition, the paper is well written and easy to follow.

Some suggestion:
(1)  The efficacy of uniformity regularizer is not clear. The paper claims “ it preserves maximal information and eliminates the feature collapse issue”.  But it would be better to understand its impact if given detailed explanation and specific examples. From the experiment table 4,  the contribution of uniformity regularizer in GARD seems marginal.

**Questions:**

(1)  what are specific features used in local semantic evolvement learning (e.g., features of parent nodes ) and features used in global semantic evolvement learning.
(2) In local semantic evolvement learning, why we need a bi-way reconstructions for both parent node and child node i? instead of one-way (e.g., child feature reconstruction from parent)
(3) in global semantic evolvement learning, the hop number of the graph determines neighboring nodes. what is the impact of hop number in graph for GRAD

**Reviewer Confidence:**

4: The reviewer is certain that the evaluation is correct and very familiar with the relevant literature

**Scope:**

4: The work is relevant to the Web and to the track, and is of broad interest to the community

---

### Official Review · Reviewer_mNwp · 2023-12-10

**Novelty:** 5
**Technical Quality:** 4

**Review:**

This paper a novel approach called the Semantic Evolvement Enhanced Graph Autoencoder for Rumor Detection (GARD) model, which addresses the challenge of detecting rumors on social media. The model incorporates semantic evolvement information of events by capturing local semantic changes and global semantic evolvement through a graph autoencoder and reconstruction strategies. By combining semantic evolvement information with propagation structure information, GARD achieves a comprehensive understanding of event propagation, enabling accurate and robust rumor detection.

The paper introduces a novel approach, the GARD model, which incorporates semantic evolvement information to enhance rumor detection on social media. This approach goes beyond traditional methods by considering the gradual transformation of semantic meaning during news propagation, capturing both local and global semantic changes.

The paper provides a high-level description of the GARD model but lacks specific implementation details. Information regarding hyperparameter tuning, architecture design choices, and computational requirements is important for reproducibility and practical implementation. Providing more detailed information would enhance the practicality and applicability of the proposed approach.

One potential weak point of this work is the lack of a comprehensive study on the model's efficiency. While the paper focuses on the effectiveness and performance of the GARD model in rumor detection, it does not thoroughly investigate the computational efficiency or resource requirements of the proposed approach. In real-world scenarios, where large-scale social media data is involved, efficiency becomes a critical factor for practical deployment. Understanding the model's computational demands, such as training time, memory usage, and inference speed, would provide valuable insights for assessing its scalability and feasibility in real-time applications.

**Questions:**

The paper should provide more detailed information about parameter settings and comparisons with baselines to enhance reproducibility and practical implementation. Additionally, a comprehensive study on the efficiency of the proposed framework is needed to assess its computational demands and scalability in real-world applications. Addressing these aspects would improve the practicality and applicability of the proposed approach.

**Reviewer Confidence:**

2: The reviewer is willing to defend the evaluation, but it is likely that the reviewer did not understand parts of the paper

**Scope:**

3: The work is somewhat relevant to the Web and to the track, and is of narrow interest to a sub-community

---

### Decision · Program_Chairs · 2024-01-22

**Decision:**

Accept

**Comment:**

The paper addresses the challenge of rumor detection on social media and proposes GARD, a Graph Autoencoder model that incorporates semantic evolvement information alongside propagation structure to achieve accurate and robust detection. GARD outperforms state-of-the-art approaches in overall performance and early rumor detection on three benchmark datasets, emphasizing the importance of capturing semantic changes during event propagation.

 + The approach is interesting to an important problem
 - The solution clearly outperforms the state of the art
 - The writing of the paper can be improved.
 - The evaluation should deep dive on why the model fails when it does.